# Variability in Skeletal Muscle Protein Synthesis Rates in Critically Ill Patients

**DOI:** 10.3390/nu14183733

**Published:** 2022-09-10

**Authors:** Inga Tjäder, Maria Klaude, Ali Ait Hssain, Christelle Guillet, Inger Nennesmo, Jan Wernerman, Olav Rooyackers

**Affiliations:** 1Department of Perioperative Medicine and Intensive Care, Karolinska University Hospital, 14186 Huddinge, Sweden; 2Department of Intensive Care Unit, University Hospital of Clermont-Ferrand, 63100 Clermont-Ferrand, France; 3Unité de Nutrition Humaine, Clermont Université, Université d’Auvergne, INRA, 63001 Clermont-Ferrand, France; 4Department of Pathology, Karolinska Institutet, 17177 Stockholm, Sweden; 5Division of Anesthesiology and Intensive Care, CLINTEC, Karolinska Institutet, 14152 Huddinge, Sweden

**Keywords:** protein synthesis, stable isotopes, muscle morphology, multiple organ failure

## Abstract

(1) Background: Muscle protein synthesis in critically ill patients is, on average, normal despite dramatic muscle loss, but the variation is much larger than in controls. Here, we evaluate if this variation is due to 1) heterogeneity in synthesis rates, 2) morphological variation or infiltrating cells, or 3) heterogeneity in the synthesis of different protein fractions. (2) Methods: Muscle biopsies were taken from both legs of critically ill patients (*n* = 17). Mixed and mitochondrial protein synthesis rates and morphologies were evaluated in both legs. Synthesis rates of myosin and actin were determined in combined biopsies and compared with controls. (3) Results: Muscle protein synthesis rates had a large variability in the patients (1.4–10.8%/day). No differences in mixed and mitochondrial protein synthesis rates between both legs were observed. A microscopic examination revealed no morphological differences between the two legs or any infiltrating inflammatory cells. The synthesis rates for myosin were lower and for actin they were higher in the muscles of critically ill patients, compared with the controls. (4) Conclusions: The large variation in muscle protein synthesis rates in critically ill patients is not the result of heterogeneity in synthesis rates, nor due to infiltrating cells. There are differences in the synthesis rates of different proteins, but these do not explain the larger variations.

## 1. Introduction

Critical illness is characterised by a dramatic and progressive loss of muscle mass during intensive care treatment. This can add up to 20% in the first 10 days of ICU treatment in the sickest patients [1]. Several studies have shown that a larger loss of muscle mass is related to a worsened outcome [2,3]. This loss often leads to intensive care unit-acquired weakness (ICUAW), but not always [4,5]. Other reasons for ICUAW are acquired myopathy or acquired neuropathy [6,7]. Both muscle wasting and weakness are not confined to locomotive muscle only, but also affect the respiratory muscles, and this is often related to prolonged mechanical ventilation [8].

The wasting of muscle mass or protein is due to an imbalance of protein synthesis and breakdown. In critically ill patients this seems to be the result of a dramatically increased muscle protein breakdown, whereas muscle protein synthesis rates are, on average, normal [1,9,10,11]. In addition, repeated determinations of the muscle protein synthesis rates in intensive care patients have shown no alteration [9,10] nor an increase [1,12] over time. The average muscle protein synthesis rate of critically ill patients is within the range of normal healthy individuals, but a larger variation is seen among the critically ill. In healthy volunteers, the mean fractional protein synthesis rate is 1.7 ± 0.3%/day (range 1.3–2.3%/day) [13], whereas in critically ill patients, the fractional synthesis rate is, on average, 1.8%/day, with a variation between 0.5 and 3.9%/day [9,10,14]. In critically ill patients, the inter-individual coefficient of variation is four-fold higher than in healthy individuals, e.g., about 40% [9,10]. The reason for this larger variation is not known. The fractional protein synthesis rates measured in both legs at the same time in the healthy volunteers showed an intra-individual coefficient of variation of 7.7% [13]. These results indicate that the muscle protein synthesis rate is homogenous and reproducible within the vastus lateralis muscle of healthy individuals. However, no studies measuring intra-individual variation in ICU patients are available.

The overall aim of this study was to describe the possible reasons for the larger variation in the muscle protein synthesis rate in ICU patients. We investigated whether the large variation was due to a high heterogeneity in the muscle protein synthesis within the same muscle from different legs, the differences in the synthesis rates of the different protein fractions in the muscles of these patients, or a high variability in morphology in the muscles of ICU patients. To investigate if the variability is a local phenomenon due to a heterogeneity within the vastus lateralis muscle, the protein fractional synthesis rates (FSRs) for mixed protein and mitochondrial protein were measured simultaneously in both legs of critically ill ICU patients. We chose to study both legs to avoid any influence of the first biopsy on the second biopsy in the same leg. In addition, the synthesis rates of the two main contractile proteins in the skeletal muscle, actin and myosin were compared with values from healthy control subjects. For the latter measurements, not enough biopsy material was available to analyse these for both legs. The FSR measures the direct incorporation of a labelled amino acid into the protein of interest, and gives the percentage (fraction) of the protein renewed per day. Finally, we determined if the infiltrating inflammatory cells could explain the large variability in synthesis rates by using them histologically.

## 2. Materials and Methods

### 2.1. Subjects

Mechanically ventilated patients (*n* = 17) who had been admitted to the multi-disciplinary intensive care unit (ICU) at Karolinska University Hospital at Huddinge were prospectively included in the study over a 15-month period (February 2002–May 2003). Patients with all diagnoses were included. Exclusion criteria were age below 18 y, severe liver failure, patients undergoing dialysis, and patients with other reasons for an impaired coagulation, or other reasons prohibiting muscle biopsies. Nutrition was given according to the routines of the unit with a target of 20–25 kcal/kg/24 h and 0.9–1.2 g protein/kg/24 h. Patients were fully fed within 48 h of their ICU arrival, with a combination of parenteral (Kabiven, Fresenius-Kabi, Uppsala, Sweden) and enteral nutrition (Fresubin, Fresenius-Kabi, Uppsala, Sweden). As a control group, 10 patients who were undergoing elective surgery at Ersta Hospital (Stockholm) were included for the measurements of actin and myosin FRSs.

The data presented here are the original purpose of the study, and all analyses were performed within 5 years of completing the patient inclusion. However, some additional measurements on mitochondria, gene expression and proteolytic enzymes, from both the critically ill patients and the controls, have been published previously [15,16]. In the present paper, we also present the mitochondrial synthesis rates from both legs to study the larger variation, although the mean values of the same results have been presented previously [15].

### 2.2. Study Protocol

#### 2.2.1. Fractional Synthesis Rates (FSRs)

Measurements were performed during the daytime when the patients were treated in the ICU. The determination of the muscle FSRs started with an intravenous injection of L-[2H5] phenylalanine (Cambridge Isotope Laboratory, Andover, MA, USA) over 10 min (45 mg/kg body weight as a 2% solution at 10 molar % excess (MPE)). Blood samples were taken before, and at 5, 10, 15, 30, 50, 70 and 90 min after the phenylalanine injection for determination of the L-[2H5] phenylalanine enrichment in the plasma. This was used as the precursor to pool enrichment.

#### 2.2.2. Muscle Biopsies

Percutaneous muscle biopsies were taken with a Bergström needle at about 90 and 92 min after the start of isotope injection, from the left and right legs, respectively [17], in the critically ill patients, and at 90 min in the controls. The biopsies were taken from the lateral portion of the quadriceps femoris muscle, 10–20 cm above the knee on both sides. Local anaesthesia was given on the skin. In the control patients, the biopsies were taken immediately after the induction of anaesthesia. The biopsy material was carefully dissected to remove visible fat and connective tissues, and then divided into two equal parts. One part, for FSR measurements, was frozen in liquid nitrogen within 60 s and stored at −80 °C pending analysis. The parts for histological examination were immediately transported to the neuropathology laboratory, where they were frozen for later analyses. The muscle specimens were blinded for the investigators performing the determinations and morphology assessments.

#### 2.2.3. Measurement of FSR

In the critically ill patients, the FSRs of the mixed and mitochondrial proteins were measured in biopsies from both legs separately for determining the variation of protein synthesis. Not enough muscle sample was left from both legs to also measure the actin and myosin FSRs separately, so the FSRs of actin and myosin were determined in combined material from the two biopsies, and only one number is presented. Furthermore, for the control patients, the biopsies were too small for all analyses, and we opted to measure the protein synthesis rates of the different protein fractions over the mixed protein synthesis, since no previous comparisons of these protein fractions between critically ill patients and controls have been performed.

The preparation of mixed and mitochondrial proteins for the measurement of their FSRs has been described in detail previously [13,15,16]. For the measurements of the myosin and actin protein synthesis rates, the skeletal muscle samples were homogenized in a buffer containing 100 mM KCL, 50 mM Tris/HCL, 5 mM MgCL_2_, 1.8 mM ATP and 1 mM EDTA (ethylenediaminetetraacetic acid). The homogenate was centrifuged at 1500× *g* at 4 °C for 10 min. The pellet contained mainly myofibrillar protein. This pellet was diluted in a buffer (0.45 M KCl, 0.2 M Mg(CH_3_COO)_2_, 1 mM EGTA (ethylene glycol-bis(β-aminoethyl ether)-N,N,N’,N’-tetraacetic acid), 1 mM DTT (Ditiotreitol), 20 mM Tris-maléate, 10 mM ATP, pH 6.8), and proteases inhibitors were added. After an incubation of 1 h at 4 °C, the pellet was centrifugated at 10,000× *g* for 15 min at 4 °C. The supernatant contained mainly myosin, which was isolated as described above [18]. The pellet contained mainly actin. This pellet was suspended in 0.8 M KCl buffer, 5 mM beta-mercaptoethanol and 0.2 M Mg(CH_3_COO)_2_ by stirring for 5 min. This homogenate was centrifuged for 15 min at 10,000× *g* at 4 °C. A solution of 2 mM HCO3- supplemented with protease inhibitors was added to the pellet. This homogenate was incubated for 1 h at 4 °C, and immediately after that it was centrifuged at 200,000× *g* for 1 h at 4 °C. The obtained supernatant contained actin, which was precipitated with 10% TCA (trichloroacetic acid). Representative protein enrichments for the myosin and actin fractions are shown in Appendix A.

The measurement of the FSR in muscle tissue by the flooding dose technique has been described in detail previously [13]. We chose to use the flooding dose technique with labelled phenylalanine, since this amino acid has a small pool size which allowed us to equilibrate all pools with the labelled phenylalanine fast, and also because this technique does not depend on a steady state for the labelled phenylalanine. In brief, the determination of L- [2H5] phenylalanine enrichment in the plasma samples, as well as in the samples of hydrolyzed muscle protein fractions, was completed by gas chromatography–mass spectrometry with electron-impact ionization and selective ion monitoring. The enrichment in plasma was measured by monitoring the ions at masses of mass/charge 336 and 341 of the tertiary butyldimethylsilyl derivate of phenylalanine. The enrichment of phenylalanine from protein hydrolysates was measured following enzymatic decarboxylation to β-phenylethylamine, and the subsequent analysis of its tertiary butyldimethylsilyl derivate at masses of mass/charge 180(M+2) and 183(M+5) [19]. This conversion was completed to allow measurements of low enrichments using a GC-MS approach [13]. The actual L- [2H5] phenylalanine enrichment in protein was calculated by comparison with a standard curve. The determinations were performed on a quadrupole mass spectrometer (Agilent 5973N, Agilent Technology, Kista, Sweden).

The fractional rate of the protein synthesis (FSR in %/day) was calculated from the formula:FSR = [P(t) − P(0)] × 100/A

P(0) and P(t) are the enrichments of phenylalanine in the muscle protein at the beginning and at the end of the incorporation period (molar percent excess), and A is the area under the curve for plasma phenylalanine enrichment (MPE × time in days). Since an L-[2H5] phenylalanine was used, and none of the subjects had received this tracer before, baseline enrichment was assumed to be zero.

#### 2.2.4. Muscle Morphology

From each biopsy, six serial eight μm thick sections were prepared using a cryostat and stained with haematoxylin–eosin. The muscle specimens were examined for signs of degeneration/regeneration of muscle fibres, atrophy, number of internal nuclei and inflammatory cell infiltration. This evaluation was completed by a trained muscle pathologist (IN) blinded for the biopsy origin.

### 2.3. Statistical Analyses

Data are presented as medians (range) when not otherwise indicated. The comparison of the results from the two legs was performed with the Wilcoxon matched-pair test (Statistica 10). For the comparison between patients and controls, the Mann–Whitney U test was used (Statistica 10). Dahlberg’s calculations were used for calculating the coefficient of variation for the measurements in the two legs [20]. Post hoc correlations were performed by regression analysis (Statistica 10).

## 3. Results

### 3.1. Patient Characteristics

Muscle protein synthesis rates and muscle morphology were studied in biopsies of both legs of critically ill patients between days 1 and 42 of their ICU stay (Appendix A. Eight women and nine men were included with a median age of 69 (range 25–77). Nine of the patients included had non-surgical backgrounds, and eight patients were treated for surgical complications. In 9/17 patients there was a failure of more than one organ system (SOFA > 2 for an individual organ; SOFA: Sequential Organ Failure Assessment). All patients were sedated with propofol together with intermittent doses of analgesics. All patients required mechanical ventilation. At the time of study, all patients were circulatory stabilised, although fourteen patients required a vasopressor and/or inotropic support. The intravenous high-dose corticosteroid betamethason was given to four patients, and another six patients received low-dose hydrocortisone or betamethason as a substitution therapy. Antibiotics were given to all patients and nine patients were treated with more than one antibiotic. Anti-fungal therapy was given to four patients. At the time of study, three of the patients had indwelling thoracic epidural catheters with a continuous infusion of local anesthetics and short-acting opioids. Total parenteral nutrition, enteral nutrition or combinations were given to 15 of the patients, and the two remaining patients received glucose 50 mg/mL and glucose 100 mg/mL, respectively. All patients received insulin intravenously to keep their blood glucose between 5 and 8 mmol/L.

The control patients underwent elective surgery for either gallbladder disease or an inguinal hernia, but they were metabolically healthy. The control subjects consisted of one woman and nine men with a median age of 70 years (range 48–76) and a median BMI of 26.7 kg/m^2^ (range 19.5–33.1).

### 3.2. Variation in Muscle Protein Synthesis Rates

All the enrichments for the labelled phenylalanine are given in Appendix A. The medium value for the mixed protein FSRs was 1.7%/d, ranging from 1.4 to 10.8%/d. The inter-individual variation was 85%. The FSRs for the individual patients showed very good agreement between the left and the right leg (Figure 1). The medium protein FSR in the left leg was 1.7%/d (1.4–10.9) and in the right leg 1.9%/d (1.4–10.7) (*p* = 0.15). The intra-individual variation was 11% between the two legs.

For mitochondrial protein FSRs the median was 2.5%/d (range 1.6–11.7%/d). The inter-individual variation was 76%. The mitochondrial FSRs were not different (*p* = 0.14) between the left leg (median 2.4%/d; range 1.5–12.0%/d) and the right leg (median 2.6%/d; range 1.4–11.3%/d). The mitochondrial FSRs for the individual patients showed very good agreement between the left and the right legs (Figure 1). The intra-individual variation for the mitochondrial protein FSRs was 17%, using the data from the two legs.

In a post hoc analysis, no statistical correlations could be found between the FSRs and the severity of clinical stress in terms of APACHE II (Acute Physiology and Chronic Health Evaluation) at admission, nor SOFA scores on the study day. The only clinical parameter that statistically correlates with the muscle protein FSRs was the number of days in the ICU (r = 0.93, *p* < 0.0001; Figure 2). Furthermore, when the two obvious outliers were excluded, the statistical correlation remained statistically significant (r = 0.73, *p* = 0.002).

### 3.3. Morphology

A microscopic examination revealed no differences between the two legs. In nine patients, there were variations in fibre size, varying from mild to the presence of small groups of atrophic fibres, indicating neurogenic damage in some patients (Appendix A. Two patients had morphological changes consistent with critical illness-acquired myopathy [21]. The remaining biopsies were considered normal. Infiltrating inflammatory cells were not present in the muscle tissue.

### 3.4. Synthesis Rates of Actin and Myosin

The FSRs of myosin were lower in the ICU patients than in the controls (0.79: 0.31–6.95%/d versus 1.17: 0.95–2.25%/d; *p* = 0.003). The FRSs of actin were higher in the ICU patients compared to the controls (3.14: 1.58–6.38%/d versus 2.44: 2.01–4.20%/d; *p* = 0.05) (Figure 3).

In the post hoc analyses, the FSRs of myosin (r = 0.73), actin (r = 0.88) and mitochondrial protein (r = 0.99) correlated with the FSRs of mixed protein (Figure 4), showing that the patients with high mixed protein FSRs had the highest FSRs of all measured proteins.

## 4. Discussion

The first aim of this study was to compare the muscle protein synthesis of the two legs in critically ill ICU patients in order to find the possible reasons for the large variability in the FSRs observed in these patients. The main result of the study was that the muscle protein FSRs in the ICU patients were similar in the two legs, ruling out heterogeneity within the same muscle type as an explanation for the large scatter. The second aim was to examine muscle morphology in specimens of the quadriceps muscles of the two legs, with the main focus of identifying infiltrating inflammatory cells as a reason for the large variation. No local differences were found between the two legs, which strongly supports the primary results that the infiltration of inflammatory cells does not explain the high variation in protein synthesis. Finally, the in vivo synthesis rates of the two main protein fractions in muscle, actin and myosin show a lower myosin synthesis rate and a higher actin synthesis rate in the critically ill patients, compared with the control patients.

ICU patients lost muscle proteins at a rate of 1.5% per day during the initial 2–3 weeks of ICU stay [9]. On a group level, this loss does not seem to be the result of a decrease in muscle protein synthesis rates, but is driven by a dramatically increased protein breakdown [11]. However, on an individual level, there is a large variation with some patients having low synthesis rates and others having very high rates. The large scatter has been observed before by our research group [9,10,14], but also by others in septic patients [1] and burn patients [22]. This higher variation has also been observed in the expression of the genes involved in protein synthesis [23]. The present results confirm the larger variation in critically ill patients, although this was not the prime objective of the study, since the patient groups and control groups were not equal in size. In the present study, we showed that this variation was not due to a large heterogeneity in muscle FSR nor the infiltration of immune cells. The intra-individual variation is 11–17% for the mixed and mitochondrial proteins, respectively, which is similar to, or slightly higher than, that observed in a study of healthy volunteers with an intra-individual variation of 8% [13]. This indicated that the variation was due to a variation in the actual muscle protein synthesis. Future studies should elucidate what the underlying reason for this large variation is, and how this is related to the outcome. Heterogeneous clinical phenotypes are likely to be part of the explanation, but the present study is too small to address this.

High synthesis rates of muscle mixed protein have been reported earlier in individual ICU patients [9,10] with FSRs of 3–4%/day. Muscle protein synthesis has, until now, been measured in a total of 73 critically ill patients, and among them 7/73 had a fractional synthesis rate above 3% per 24 h. The upper level in healthy volunteers is 2.3%/day (mean + 2 × SD) [13]. The critically ill patients were not a homogenous group, and all patients had been exposed to various degrees of metabolic stress. In a post hoc analysis, no statistical correlations could be found between the muscle protein synthesis rates and the severity of clinical stress in terms of APACHE II at admission, nor SOFA score on the study day. The only clinical parameter that statistically correlated with the synthesis rates was the number of days in the ICU. Although the results from this post hoc analysis are hypothesis-generating only, they suggest that muscle protein synthesis increases with extended ICU stay, or perhaps with survival in the ICU. In already published studies, two muscle measurements of muscle protein synthesis five days apart have been performed in a total of 73 patients [9,10,24]. Analyses of all patients together have shown no statistical difference in muscle protein synthesis rates within the five-day study period (1.5 (1.2–1.9) versus 1.6 (1.2–2.4)%/day; *p* = 0.16). In a recent study, muscle protein synthesis, measured as FSR, was significantly higher in critically ill patients on day 7 compared with day 1 [1]. However, in the same patients, when muscle protein synthesis rates were measured with another method (arterio-venous tracer balance), this was not observed. We have, in a more recent study, measured muscle protein turnover, using the arterio-venous tracer balance method in ICU patients between 10 and 40 days of ICU treatment [12]. This prospective study has clearly shown that muscle protein synthesis rates increase with the length of stay in the ICU. At the same time, muscle protein breakdown rates are high in these patients and do not change over time.

The relatively low intra-individual coefficient of variation (11%) for muscle FSR reported in this study, compared to the high inter-individual variation of about 80%, demonstrates that changes in muscle FSR may also be detected in small groups of patients, but only when paired sampling in the same patients is performed. Ten patients are enough to show a 10% difference in the fractional synthesis rates, with a power of 0.8 in a paired experiment, whereas > 400 are needed if two groups of patients are compared.

One explanation for the extremely high variation in synthesis rates could be that a different set of proteins have been synthesised, or even that the muscle produces poly-peptides that are never finished into intact proteins. However, when synthesis rates of different muscle protein fractions are measured, this is not supported. The different protein fractions respond slightly differently to critical illness, with a lower synthesis rate for myosin and a higher rate for actin. Mitochondrial protein synthesis rates, on average, are not different from the controls [15]. Previous studies have shown that patients with critical illness myopathy have a decreased ratio of myosin to actin in comparison with the controls [21,25]. This could well be the result of the decreased synthesis rate of myosin and the increased synthesis rate of actin that we observed in the present study.

Furthermore, for these different fractions, it is obvious that the variation for the critically ill patients is larger than for the controls. When the synthesis rates of these proteins were correlated to the mixed protein synthesis rates (Figure 4), they correlated well. The lowest statistical correlation was observed for myosin and the highest for mitochondrial protein. These results show that the higher synthesis rates of mixed muscle protein are due to an increased synthesis of real muscle proteins, and not different proteins or even incomplete polypeptides. The close correlations with synthesis rates of the different protein fractions also indicated that the large variation in mixed muscle protein synthesis rates is most likely not due to a variation in the synthesis of specific proteins.

A limitation of the study is that we only measured fractional synthesis rates and not absolute synthesis rates. However, accurate assessments of the muscle protein mass are not possible in critically ill patients. Another limitation is that a heterogeneous group of ICU patients was included, which could explain the large variation. However, none of the clinical parameters tested (SOFA, APACHE) correlated with a variation in the muscle protein synthesis rates, except for the length of stay in the ICU.

## 5. Conclusions

Muscle protein synthesis rates in critically ill patients were demonstrated to be homogenous and reproducible in both legs, as was the histological picture of the skeletal muscle. This ruled out heterogeneity and artefacts as explanations for the large scatter in the muscle protein synthesis rates reported earlier and confirmed in this study. Analyses of the synthesis rates of different muscle proteins showed that not all proteins respond similarly, with lower myosin synthesis rates and higher synthesis rates of actin, and that these differences are not likely an explanation of the large variation in mixed protein synthesis.

## Figures and Tables

**Figure 1 nutrients-14-03733-f001:**
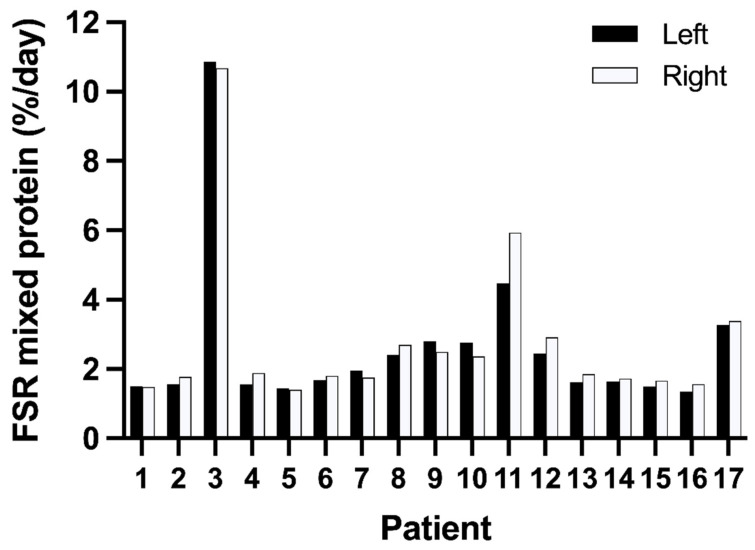
Fractional synthesis rates (FSRs) in skeletal muscle mixed protein (**top**) and mitochondrial protein (**bottom**) of both legs of critically ill patients (*n* = 17). Individual values are given where open bars represent the left leg and filled bars represent the right leg. Mean values of mitochondrial FSR have been published previously [15].

**Figure 2 nutrients-14-03733-f002:**
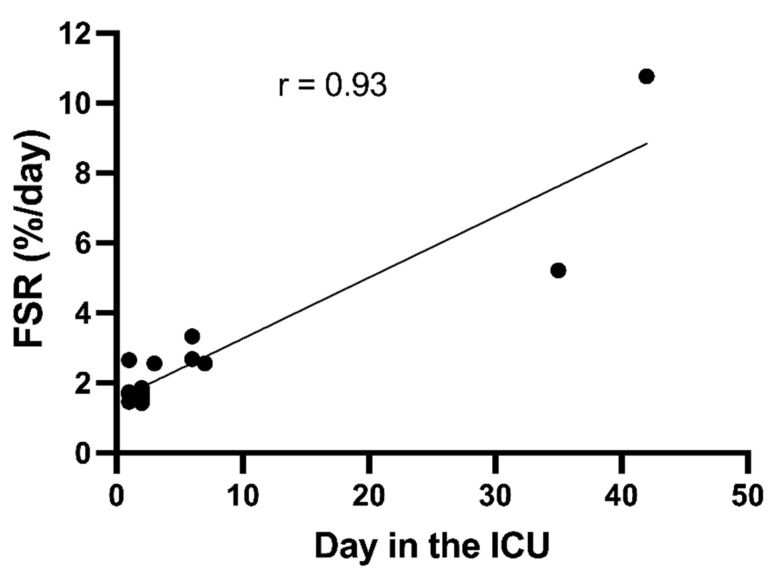
Correlation between mean in the fractional synthesis rates (FSRs) of skeletal muscle mixed protein and the days of ICU stay when patients were studied (*n* = 17) using regression analysis (**top**). The correlation was also calculated with the two obvious long-stayers excluded (**bottom**).

**Figure 3 nutrients-14-03733-f003:**
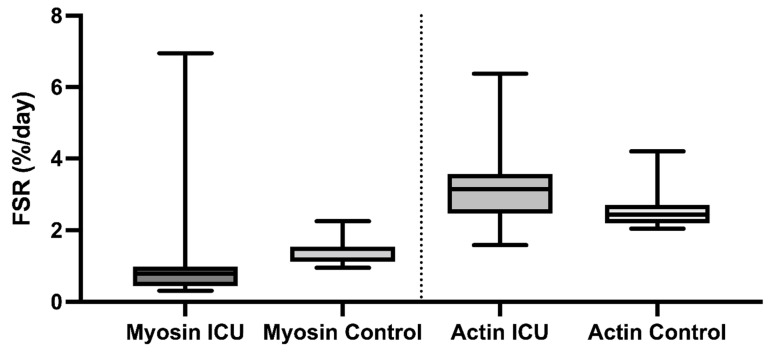
Fractional synthesis rates (FSRs) of myosin and actin from skeletal muscle in critically ill patients (ICU) and healthy controls (patients undergoing elective minor surgery) measured in only one leg. Individual values are given as dots and median values by the lines. Groups were compared using a Mann–Whitney U test.

**Figure 4 nutrients-14-03733-f004:**
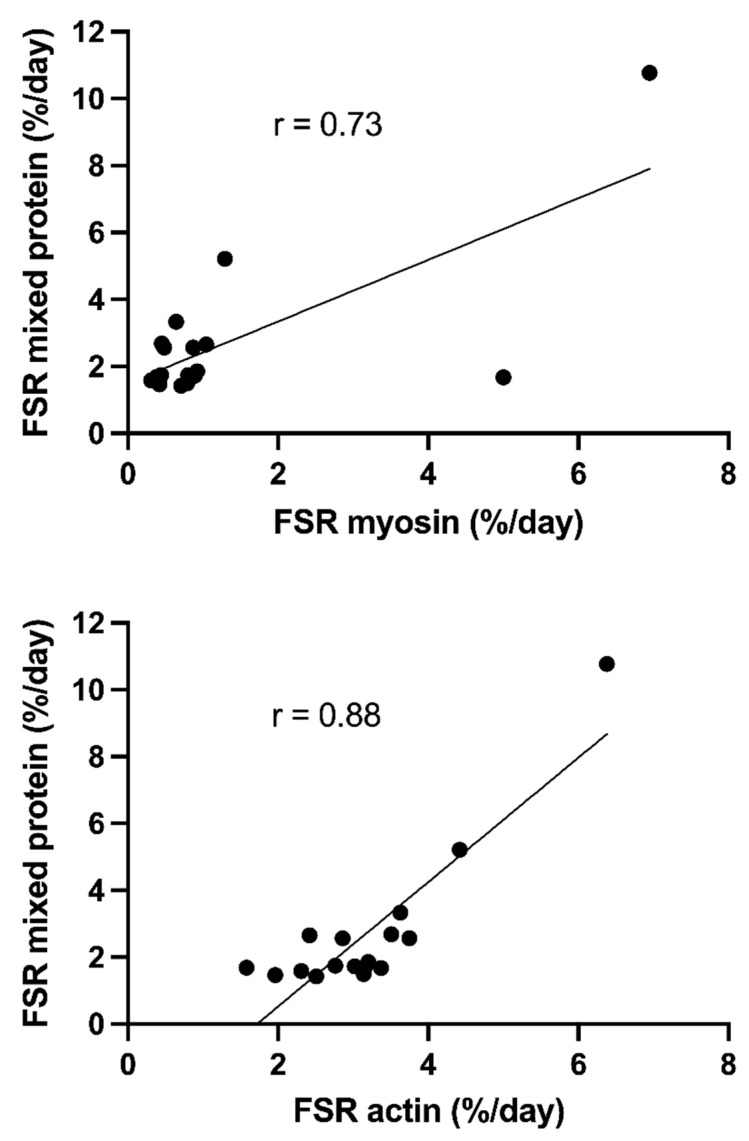
Correlation between means in the fractional synthesis rates (FSRs) of skeletal muscle mixed protein and FSRs of 3 protein fractions (top: myosin; middle: actin; bottom: mitochondrial protein) (*n* = 17) using regression analysis. The mitochondrial FSRs have been previously published [15].

## Data Availability

Data for the measurements of protein synthesis are in the Appendix A. More clinical data can be shared on request to the corresponding author.

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
