# Peer review of "Variability in Skeletal Muscle Protein Synthesis Rates in Critically Ill Patients"

_nutrients, 2022, doi:10.3390/nu14183733_

Round 1
Reviewer 1 Report
The manuscript by Tjader et al examined whether variability in muscle protein synthesis in ICU patients could be explained by differences in rates of synthesis in contralateral legs, or by differences in rates of synthesis of specific proteins (mitochondrial, myosin, actin). Some of the data are derived from works previously published. An understanding of regulation of muscle protein synthesis in ICU patients may help to limit/prevent muscle loss in these patients. So, in principle the manuscript has merits. However, in the introduction section, the authors acknowledged that much of the muscle loss in ICU patients is likely driven by augmented proteolysis as compared to changes in protein synthesis for which there appears to be less consistent patterns. Given this, it is not clear why the authors chose to focus on protein synthesis rather than on proteolysis. Some of the data presented have been published elsewhere (L91-7). The variability in protein synthesis referred to was not really shown in the manuscript except in Fig 3 where they showed data from control and ICU patients. For this figure, much of the variability observed in ICU patients compared to control is likely driven by the bigger n and the heterogeneity in patient population for the ICU compared to control, which is also made up of predominantly male participants (9:1 vs 9:8 in ICU patients). For these reasons, it is not entirely clear what the data presented add. Because the medical conditions the patients were admitted into the ICU for had nothing to do with injuries or differential treatments/injuries to the limbs, the rationale for expecting differences in muscle protein synthesis between the limbs is not clear.
L48, 56-57: critically ill patients and divergent rates of muscle protein synthesis. Isn’t this to be expected given the fact that the patients were admitted into the ICU for divergent/myriads of health issues?
L83-85: What are the reasons for these exclusions? It is already quite an heterogenous group: BMI, length of stay in ICU, nutritional support and type, corticosteroid admin, etc.
L111-3: It is stated that in the control patients, muscle sample was done close to initiation of anesthesia, but it is not clear when sampling was done in the ICU patients relative to the initiation of anesthesia, since this in itself potentially can affect protein synthesis. With regards to the correlation data of protein synthesis to length of stay in ICU, were the subjects studied after the same length of stay in the ICU?
In L150-8, it is not clear why phenylalanine enrichment was used for the determination of plasma isotopic enrichment but phenethylamine was used to determine isotope enrichment in protein-bound phenylalanine.
L169-71: it is not clear how muscle morphology was examined. Related, no data are presented on the measures mentioned here (L235-40). Also see L263-267.
L308-12: Not sure of the relevance of this. Perhaps in exercise model where one leg is exercised and the other serves as the within-patient control. But in ICU patients with systemic perturbations, why would one be comparing one leg to the other within the same patients?
L313- and L341-2: As mentioned before, comparison between control and ICU patients was only barely done in this manuscript.
Quality of some of the figures could be improved. In particular, Fig 2-4 look blurred.
MINOR:
Please provide definitions of abbreviations and explanation of other less common terms/words (APACHE II, SOFA, etc.) used for the benefit of those that might not be familiar with such terms. Also please see terms/words used in Supp Table 1 need to be explained. Quality of Supp Table 2 is suboptimal: looks blurred.
L77: please check grammar
Author Response
We like to thank the reviewers for their thorough review of our manuscript and have incorporated most of the comments. Please see detailed relies below.
Reply to Reviewer 1
The manuscript by Tjader et al examined whether variability in muscle protein synthesis in ICU patients could be explained by differences in rates of synthesis in contralateral legs, or by differences in rates of synthesis of specific proteins (mitochondrial, myosin, actin). Some of the data are derived from works previously published. An understanding of regulation of muscle protein synthesis in ICU patients may help to limit/prevent muscle loss in these patients. So, in principle the manuscript has merits. However, in the introduction section, the authors acknowledged that much of the muscle loss in ICU patients is likely driven by augmented proteolysis as compared to changes in protein synthesis for which there appears to be less consistent patterns. Given this, it is not clear why the authors chose to focus on protein synthesis rather than on proteolysis. Some of the data presented have been published elsewhere (L91-7). The variability in protein synthesis referred to was not really shown in the manuscript except in Fig 3 where they showed data from control and ICU patients. For this figure, much of the variability observed in ICU patients compared to control is likely driven by the bigger n and the heterogeneity in patient population for the ICU compared to control, which is also made up of predominantly male participants (9:1 vs 9:8 in ICU patients). For these reasons, it is not entirely clear what the data presented add. Because the medical conditions the patients were admitted into the ICU for had nothing to do with injuries or differential treatments/injuries to the limbs, the rationale for expecting differences in muscle protein synthesis between the limbs is not clear.
Reply: We like to thank the reviewer for the comments. As the reviewer point out it is important to know the mechanism behind the severe muscle wasting of these patients to prevent or limit the loss. We therefore think that both sites (breakdown as well as synthesis) need to be investigated to understand the changes. We have focussed on breakdown in other papers and here focus on the synthesis. What we relay like to understand is why some of the critically ill patients rae increasing synthesis more than others. But before we can do this, we needed to be sure that what we are measuring and especially the large variation is representing real muscle protein synthesis and is not an artefact. This is what this paper aims to do. The larger variation in synthesis rates is not only shown in our cohorts but also in other studies. We do not believe that the larger variation shown in figure 3 is due to a larger ICU cohort since all other studies comparing ICU and controls show a larger variation in the ICU groups and also muscle protein synthesis rates of 5 or 10%/day have never been reported in healthy subjects in any situation. We have updated the second paragraph in the discussion addressing this. For the question whether there possibly is a local heterogeneity in muscle protein synthesis, we opted for studying both legs rather than taking 2 biopsies from the same leg with the possible influence or the first biopsy on the second one. We explain this now better in the Introduction.
L48, 56-57: critically ill patients and divergent rates of muscle protein synthesis. Isn’t this to be expected given the fact that the patients were admitted into the ICU for divergent/myriads of health issues?
Reply: Possibly if we had observed a large variation in decreased protein synthesis. However, the large variation with very high protein synthesis in some patients is very unexpected since all other wasting diseases are always accompanied with a lower muscle protein synthesis. So, we do not believe that the variation is due to different health issues in these patients, but this need to be proven in the future. Our aim for the present study was to make sure that what we are measuring is real muscle protein synthesis and not an artefact. After this we can investigate the reason for the variation which among others could be underlying diseases.
L83-85: What are the reasons for these exclusions? It is already quite an heterogenous group: BMI, length of stay in ICU, nutritional support and type, corticosteroid admin, etc.
Reply: All these exclusions (except the age exclusion) are related to an increased risk for coagulation problems and bleeding following the muscle biopsy. For this reason, these patients were excluded. We have clarified this in the text.
L111-3: It is stated that in the control patients, muscle sample was done close to initiation of anesthesia, but it is not clear when sampling was done in the ICU patients relative to the initiation of anesthesia, since this in itself potentially can affect protein synthesis. With regards to the correlation data of protein synthesis to length of stay in ICU, were the subjects studied after the same length of stay in the ICU?
Reply: ICU patient at the time of the study were in general sedated for most of their ICU stay (see description of patients in the Results. Patients were studies at different LOS in the ICU. The reason for this is that patients were studied when there was a possibility to perform the muscle biopsy based on both the clinical state of the patient and the availability of the research personnel. The exact LOS at the time of measurement is shown in the table in the supplement.
In L150-8, it is not clear why phenylalanine enrichment was used for the determination of plasma isotopic enrichment but phenethylamine was used to determine isotope enrichment in protein-bound phenylalanine.
Reply: This is a standard procedure that allows low enrichment to be measured using a GC-MS. By using the ex vivo conversion of phenylalanine from the protein hydrolysate to Beta-phenylethylamine, one can extract the Beta-phenylethyl from all other amino acids and basically reduce the baseline on the analysis (noise) to zero, allowing for a more accurate measurement of the low phenylalanine enrichment in the muscle protein. Since this is a standard procedure, we have not included this in the methods. We have however added a short sentence with a reference addressing this.
L169-71: it is not clear how muscle morphology was examined. Related, no data are presented on the measures mentioned here (L235-40). Also see L263-267.
Reply: We agree with reviewer that this is not clear. We have clarified this and also added a result table from these analysis in the supplement.
L308-12: Not sure of the relevance of this. Perhaps in exercise model where one leg is exercised and the other serves as the within-patient control. But in ICU patients with systemic perturbations, why would one be comparing one leg to the other within the same patients?
Reply: We basically wanted to measure muscle protein synthesis in 2 different muscle biopsies to make sure the large variation is due to a high heterogeneity in local muscle protein synthesis in the patients either due to actual differences or artefacts. We could have opted to take 2 biopsies close to each other in the same leg but didn’t want to risk that the first biopsy due to the trauma would affect the second biopsy and therefore took biopsies from the same muscle group in both legs.
L313- and L341-2: As mentioned before, comparison between control and ICU patients was only barely done in this manuscript.
Reply: We agree with the reviewer, but this was due to the fact that the controls were included with main purpose to compare the synthesis rates of the different protein fractions since this has not been done before and the small size of the biopsies of the controls didn’t allow for all the analyses to be done. We therefore used all the material for the main purpose to compare the synthesis rates of the myosin, actin and mitochondria and have unfortunately no material left for the mixed protein synthesis in the controls for comparison. We have clarified this in the method section better now.
Quality of some of the figures could be improved. In particular, Fig 2-4 look blurred.
Reply: Quality of figures has been improved.
MINOR:
Please provide definitions of abbreviations and explanation of other less common terms/words (APACHE II, SOFA, etc.) used for the benefit of those that might not be familiar with such terms. Also please see terms/words used in Supp Table 1 need to be explained. Quality of Supp Table 2 is suboptimal: looks blurred.
Reply: We like to thank the reviewer for noticing this. We have explained all the abbreviations in text and Table 1. We also replaced Table 2 with better version.
L77: please check grammar
Reply: We have changed this sentence.
Reviewer 2 Report
The manuscript entitled Variability in skeletal muscle protein synthesis rates in critically ill patients by Tjäder et al investigates the source of variability of skeletal muscle protein synthesis that has been observed in critical ill patients, while the averaged value remains insignificant different from control samples. In total, 17 patients and some control muscle biopsies were analyzed. The authors determined the inter-subject variation by determining the FSR in left and right leg, which indicated that the technical variation was quite low and the source of variation is likely introduced by the subject or subject’s diseases. Next the authors found that the only clinical parameter that correlated with the detected FSR was the time that the patients spent in the ICU. While the study is of potential interest to researchers accessing the FSR and the readers of Nutrients, I have several major points that should be addressed before I can recommend it for publication.
Major points
· It would be advisable to explain to the audience why such a heterogenous group of patients would not show a high variation in the FSR, due different background, ages etc. I believe that the individuals in the control group do show a more homogenous background. Would the source of variation not simply be explained by the sample’s background itself?
· Figure 3 shows the FSR measured for Actinin and Myosin in control and patient groups, but this does not show the variation. It is advisable to show the data in a boxplot/violin plot rather than a dot-plot hence this will give an idea about the distribution (e.g. variation) of the data. The statistical test should be used to compare the distributions not the means.
· I cannot find any data/ representing image regarding the Morphology.
· Please provide a western blot analysis to show that the measured FSR in the actinin and myosin fractions are significantly enriched.
Minor points
· Line 208 – medium should likely read median
· Line 325 ‘figure 4’ should likely read Figure 4
· Please provide some of real measurement values over time on which the FSR was estimated.
Author Response
We like to thank the reviewers for their thorough review of our manuscript and have incorporated most of the comments. Please see detailed relies below.
Reply to Reviewer 2
The manuscript entitled Variability in skeletal muscle protein synthesis rates in critically ill patients by Tjäder et al investigates the source of variability of skeletal muscle protein synthesis that has been observed in critical ill patients, while the averaged value remains insignificant different from control samples. In total, 17 patients and some control muscle biopsies were analyzed. The authors determined the inter-subject variation by determining the FSR in left and right leg, which indicated that the technical variation was quite low and the source of variation is likely introduced by the subject or subject’s diseases. Next the authors found that the only clinical parameter that correlated with the detected FSR was the time that the patients spent in the ICU. While the study is of potential interest to researchers accessing the FSR and the readers of Nutrients, I have several major points that should be addressed before I can recommend it for publication.
Major points
- It would be advisable to explain to the audience why such a heterogenous group of patients would not show a high variation in the FSR, due different background, ages etc. I believe that the individuals in the control group do show a more homogenous background. Would the source of variation not simply be explained by the sample’s background itself?
Reply: This is certainly possible but needs to be investigated and proven to be true. The primary aim of this study was to ensure that the large variation is actually a variation in muscle protein synthesis and not an artefact. After this we can start to explore the clinical/physiological reason for the larger variation which could be due to the different backgrounds and insults of the patients, but this needs to be seen. Please also see the response to Reviewer 1 on this point. We have updated the second paragraph in the discussion addressing this issue.
- Figure 3 shows the FSR measured for Actinin and Myosin in control and patient groups, but this does not show the variation. It is advisable to show the data in a boxplot/violin plot rather than a dot-plot hence this will give an idea about the distribution (e.g. variation) of the data. The statistical test should be used to compare the distributions not the means.
Reply: We have changed the graph to box plots. For the statistical test we are not sure what the reviewer wants. The data is clearly not normally distributed which allows us to use non-parametric analyses only. We consulate with our statistical advisor and were advised to stick with the nonparametric test we use now.
- I cannot find any data/ representing image regarding the Morphology.
Reply: We have now added the scoring for the morphology in the supplement.
- Please provide a western blot analysis to show that the measured FSR in the actinin and myosin fractions are significantly enriched.
Reply: We have added SDS gel electrophoresis results of the isolation in the supplementary material.
Minor points
- Line 208 – medium should likely read median
- Line 325 ‘figure 4’ should likely read Figure 4
- Please provide some of real measurement values over time on which the FSR was estimated.
Reply: We have corrected the minor points. The actual enrichments for the FRS values are given in the supplementary material.
Round 2
Reviewer 2 Report
The authors have responded to my points and I can recommend the paper for publication.